# A low-cost paper-based synthetic biology platform for analyzing gut microbiota and host biomarkers

Melissa K. Takahashi [1], Xiao Tan[1,2,3,4,5], Aaron J. Dy [1,5,6], Dana Braff[1,7], Reid T. Akana[6],
Yoshikazu Furuta [1,8], Nina Donghia[4], Ashwin Ananthakrishnan[2,3] & James J. Collins[1,4,5,6,9,10]

There is a need for large-scale, longitudinal studies to determine the mechanisms by which the gut microbiome and its interactions with the host affect human health and disease. Current methods for profiling the microbiome typically utilize next-generation sequencing applications that are expensive, slow, and complex. Here, we present a synthetic biology platform for affordable, on-demand, and simple analysis of microbiome samples using RNA toehold switch sensors in paper-based, cell-free reactions. We demonstrate species-specific detection of mRNAs from 10 different bacteria that affect human health and four clinically relevant host biomarkers. We develop a method to quantify mRNA using our toehold sensors and validate our platform on clinical stool samples by comparison to RT-qPCR. We further highlight the potential clinical utility of the platform by showing that it can be used to rapidly and inexpensively detect toxin mRNA in the diagnosis of *Clostridium difficile* infections.

[1] Institute for Medical Engineering and Science, Massachusetts Institute of Technology, 77 Massachusetts Ave, Cambridge, MA 02139, USA. [2] Division of Gastroenterology, Massachusetts General Hospital, 55 Fruit Street, Boston, MA 02114, USA. [3] Harvard Medical School, 25 Shattuck St, Boston, MA 02115, USA. [4] Wyss Institute for Biologically Inspired Engineering, Harvard University, 3 Blackfan Circle, Boston, MA 02115, USA. [5] Broad Institute of MIT and Harvard, 415 Main St, Cambridge, MA 02142, USA. [6] Department of Biological Engineering, Massachusetts Institute of Technology, 77 Massachusetts Ave, Cambridge, MA 02139, USA. [7] Department of Biomedical Engineering, Boston University, 44 Cummington Mall, Boston, MA 02215, USA. [8] Division of Infection and Immunity, Research Center for Zoonosis Control, Hokkaido University, North 20, West 10 Kita-ku, Sapporo 001-0020, Japan. [9] Synthetic Biology Center, Massachusetts Institute of Technology, 77 Massachusetts Ave, Cambridge, MA 02139, USA. [10] Harvard-MIT Program in Health Sciences and Technology, 77 Massachusetts Ave, Cambridge, MA 02139, USA. These authors contributed equally: Melissa K. Takahashi, Xiao Tan, Aaron J. Dy. Correspondence and requests for materials should be addressed to J.J.C. (email: jimjc@mit.edu)

The gut microbiome is an essential contributor to numerous processes in human health and disease, including proper development of the immune system[1], host responses to acute and chronic infections[2,3], cardiovascular disease[4], and drug metabolism[5]. It is also an important modulator of gastrointestinal function, including inflammatory bowel disease (IBD)[6,7], childhood malnutrition[8,9], and cancer immunotherapy treatment[10,11]. Increasing evidence suggests that host–microbiome interactions also play a key role in these health conditions[12–14]. Despite the progress made in our understanding of the overall gut microbiome and the roles of individual species, large-scale longitudinal studies are needed to more directly investigate the causal relationship between microbial and host changes during disease states and responses to treatment. Current methods for profiling the gut microbiome typically involve deep sequencing coupled with high-throughput bioinformatics. These techniques are expensive, slow, and require significant technical expertise to design, run, and interpret. To reduce costs, researchers often batch samples for sequencing, which can lead to significant increases in turnaround time. These limitations have severely restricted the large-scale prospective monitoring of patient cohorts that is necessary to provide more granular data on microbial changes and human health[15]. Here we present a synthetic biology platform that addresses the need for affordable, on-demand, and simple analysis of microbiome samples that can aid in monitoring large-scale patient cohorts.

Our lab has developed a paper-based diagnostic platform for portable, low-cost detection of biologically relevant RNAs[16,17]. The platform is comprised of two synthetic biology technologies. The first technology is a molecular sensor called an RNA toehold switch that can be designed to bind and detect virtually any RNA sequence[18]. The second is an in vitro cell-free transcription–translation system that is freeze-dried onto paper disks for stable, long-term storage at room temperature[16]; upon rehydration, the cell-free system can execute any genetic circuit. We combined these two technologies to form an abiotic platform for rapid and inexpensive development and deployment of biological sensors. Recently, we reduced the limit of detection of this platform to three femtomolar (fM) by adding an isothermal RNA amplification step called NASBA (nucleic acid sequence based amplification)[17]. We demonstrated the utility of our platform in detecting the presence or absence of clinically relevant RNAs, including those of Ebola[16] and Zika[17] viruses, but we were not able to quantify their concentrations.

Here, we address the need for affordable, on-demand, and simple analysis of microbiome samples by advancing our paper-based diagnostic platform for use as a research tool to quantify bacterial and host RNAs from stool samples (Fig. 1). To demonstrate the widespread applicability of our diagnostic platform, we select a panel of 10 bacteria relevant to diverse microbiome research studies. We first design toehold switch sensors that detect the V3 hypervariable region of the 16S ribosomal RNA (rRNA) for each species to mimic the standard method of identifying bacterial species through 16S ribosomal DNA sequencing. We then improve the specificity of detection by designing toehold switch sensors for species-specific mRNAs from each bacterial species, and demonstrate sensor orthogonality. Next, we develop a method that semi-quantitatively measures the concentrations of target RNAs using NASBA and toehold switch sensors, and validate this method against quantitative reverse transcription PCR (RT-qPCR) of clinical stool samples. We then develop toehold switch sensors to detect four host biomarkers, one of which, calprotectin, is well-established in clinical use[19], and another, oncostatin M (OSM), which may have an immediate impact on clinical decision-making in the treatment of IBD[20]. We validate our method against RT-qPCR using clinical samples from patients with IBD. Finally, we demonstrate an additional potential clinical application of our RNA detection platform using the example of Clostridium difficile infection (CDI), where differentiating active infection from passive colonization has been fraught with difficulty[21]. Our method shows markedly different toxin mRNA expression levels in two toxigenic C. difficile strains that would otherwise be indistinguishable by standard DNA-based qPCR diagnosis.

## Results

**Development of toehold switch sensors to detect 16S rRNA.** Toehold switch sensors are synthetic riboregulators that control the translation of a gene via RNA–RNA interactions. They utilize a designed hairpin structure to block gene translation in cis by sequestration of the ribosome binding site (RBS) and start codon. Translation is activated upon the binding of a trans-acting trigger RNA to the toehold region of the switch, which relieves the sequestration of the RBS and allows translation of the downstream gene (Fig. 2a)[18]. Toehold switch sensors can be designed to bind nearly any RNA sequence.

We designed toehold switch sensors for a panel of 10 bacteria chosen for their relevance to IBD[22,23], childhood malnutrition[8,9], and cancer immunotherapy[10,11]. To start, we targeted the 16S rRNA, because 16S rDNA profiling is a standard method for identifying bacterial species and rRNA is present at high copy numbers in bacteria. We used the series B toehold switch design from Pardee et al.[17] and the Nucleic Acids Package (NUPACK)[24] to design toehold switch sensors that target the V3 hypervariable region of the 16S rRNA for each target species. The candidate sensors were constructed to regulate the expression of the GFPmut3b gene[25], and tested using in vitro transcribed trigger RNAs (36 nucleotides) in paper-based, cell-free reactions (Supplementary Fig. 1). The best performing sensor for each species (Fig. 2b, Supplementary Data 1) was chosen based on the lowest background GFP expression in reactions with sensor alone and highest fold activation in reactions with sensors activated by cognate trigger RNA.

An individual bacterial species can comprise 1% or less of the total bacterial population within a human gut microbiome, so even highly abundant rRNA from an individual species can constitute 1–10 nanomolar (nM) RNA. Thus, unprocessed rRNA from stool samples is beyond the limit of detection of toehold switch sensors alone, which is approximately 10–30 nM[17]. We therefore incorporated NASBA, an isothermal RNA amplification technique[26], into our sample processing steps prior to detection by toehold switch sensors to improve assay sensitivity. Briefly, NASBA begins with primer-directed reverse transcription of the template RNA, which creates an RNA/DNA duplex. The template RNA strand is then degraded by RNaseH, which allows a second primer containing the T7 promoter to bind and initiate double-stranded DNA synthesis. The double-stranded DNA serves as a template for T7-mediated transcription of the target RNA. Each newly synthesized RNA strand can serve as starting material for further amplification cycles (Fig. 2c)[26] and can also be detected by the toehold sensors. We have previously shown that NASBA allows for the detection of single femtomolar concentrations of RNA[17] using toehold switch sensors in paper-based reactions.

NASBA primers were designed to amplify the V3 hypervariable region of the 16S rRNA for E. coli. We first tested the standard universal primer set routinely used to amplify the V3 region from 16S rDNA[27] for sequencing applications. We used total RNA extracted from an E. coli monoculture to screen the primers. NASBA reactions were performed for 90 min on 1 ng of total RNA and then applied to paper-based reactions containing the E. coli 16S toehold switch sensor. Unexpectedly, these primers were

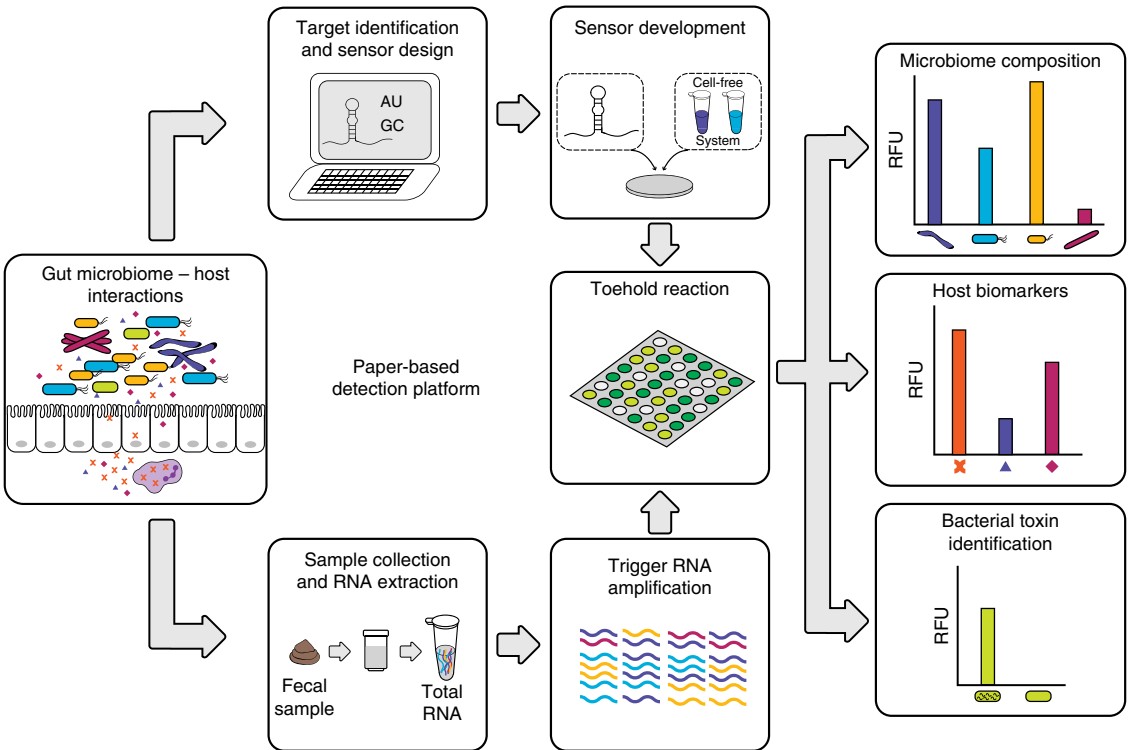

**Fig. 1** Workflow for analysis of microbiome samples using our paper-based detection platform. Once key bacteria or mRNA targets have been identified, RNA toehold switch sensors and primers for isothermal RNA amplification are designed in silico. Sensors and primers are then rapidly assembled and validated in paper-based reactions. For subsequent use, total RNA is extracted from human fecal samples using a commercially available kit. Specific RNAs are amplified via NASBA (nucleic acid sequence based amplification) and quantified using arrays of toehold switch sensors in paper-based reactions. Microbial and host biomarker RNA concentrations of the samples are determined using a simple calibration curve

not able to amplify the 16S V3 region from total RNA (Fig. 2d–E.c. 1). In order to investigate why the universal primer set performed poorly, we mapped the primer locations to chemical structure probing data for *E. coli* 30S ribosomal subunits[28] and found that the forward primer targeted nucleotides that were not structurally accessible (Supplementary Fig 2). Using the 16S rRNA structure data, we designed new NASBA primer sets and screened for the highest activation of toehold switch sensors (primer set 4). We then designed and screened NASBA primers for the other nine species using the same methodology (Fig. 2d).

We next investigated the specificity of our 16S toehold switch sensors. We synthesized trigger RNAs for each species representing the sequence that would be amplified by the NASBA primers (72–171 nucleotides) and measured the activation of each sensor when challenged with each of the 10 trigger RNAs (Fig. 2e, Supplementary Fig. 3, Supplementary Data 2). We observed good specificity for most of the 16S sensors; however, there was significant crosstalk among closely related bacteria. In the case of three closely related *Bifidobacteria*, the toehold switch sensors preferentially activate in the presence of their cognate trigger RNAs, but show significant crosstalk since the trigger sequences only differ by a few nucleotides. We also observed significant crosstalk between the *C. difficile* sensor and the trigger RNAs for *E. rectale* and *F. prausnitzii*. Although the *C. difficile* sensor is not activated by the exact 36 nucleotide triggers for *E. rectale* and *F. prausnitzii* sensors (Supplementary Fig. 4a), alignment of the NASBA-amplified RNA sequences for the three species showed that the extended sequence that is amplified by the *E. rectale* and *F. prausnitzii* NASBA primers aligned with the toehold region of the *C. difficile* sensor (Supplementary Fig. 4b, c). The 16S sensors can be used to identify and differentiate closely related families of

bacteria, but due to crosstalk, they are not suitable for discriminating among highly related bacterial species.

**Bioinformatic analysis for species-specific identification.** To address the specificity limitations of the 16S sensors, we devised a bioinformatic pipeline to identify mRNAs that are unique to any given bacterial species (Fig. 3a). Our pipeline uses the phylogenetic assignment tools Metaphlan and Metaphlan2[29] to identify a set of unique sequences for a given bacterial species. These sequences are then evaluated using a series of BLAST[30] alignments to determine the most specific markers with the highest expression in human stool (see Methods).

We followed the same steps described for 16S rRNA sensor development to develop sensors for the species-specific mRNAs. We tested candidate toehold switch sensors in paper-based reactions and selected the best performing sensor for each species (Supplementary Fig. 5). We then designed NASBA primers and screened them on total RNA extracted from monocultures for each species. The best performing NASBA primer sets were chosen based on the ability of the amplified RNA to activate the corresponding toehold switch sensor (Fig. 3b). We note the apparent variation in the efficiency of amplification between species and attribute this to the variation in abundance of the mRNAs in each total RNA sample and possible differences in the structural accessibility of these transcripts. Finally, we tested the specificity of our toehold switch sensors by synthesizing trigger RNAs for each species representing the sequence that would be amplified by the NASBA primers and tested each sensor against each of the 10 trigger RNAs. We observed greatly improved sensor specificity compared to our 16S sensors with no

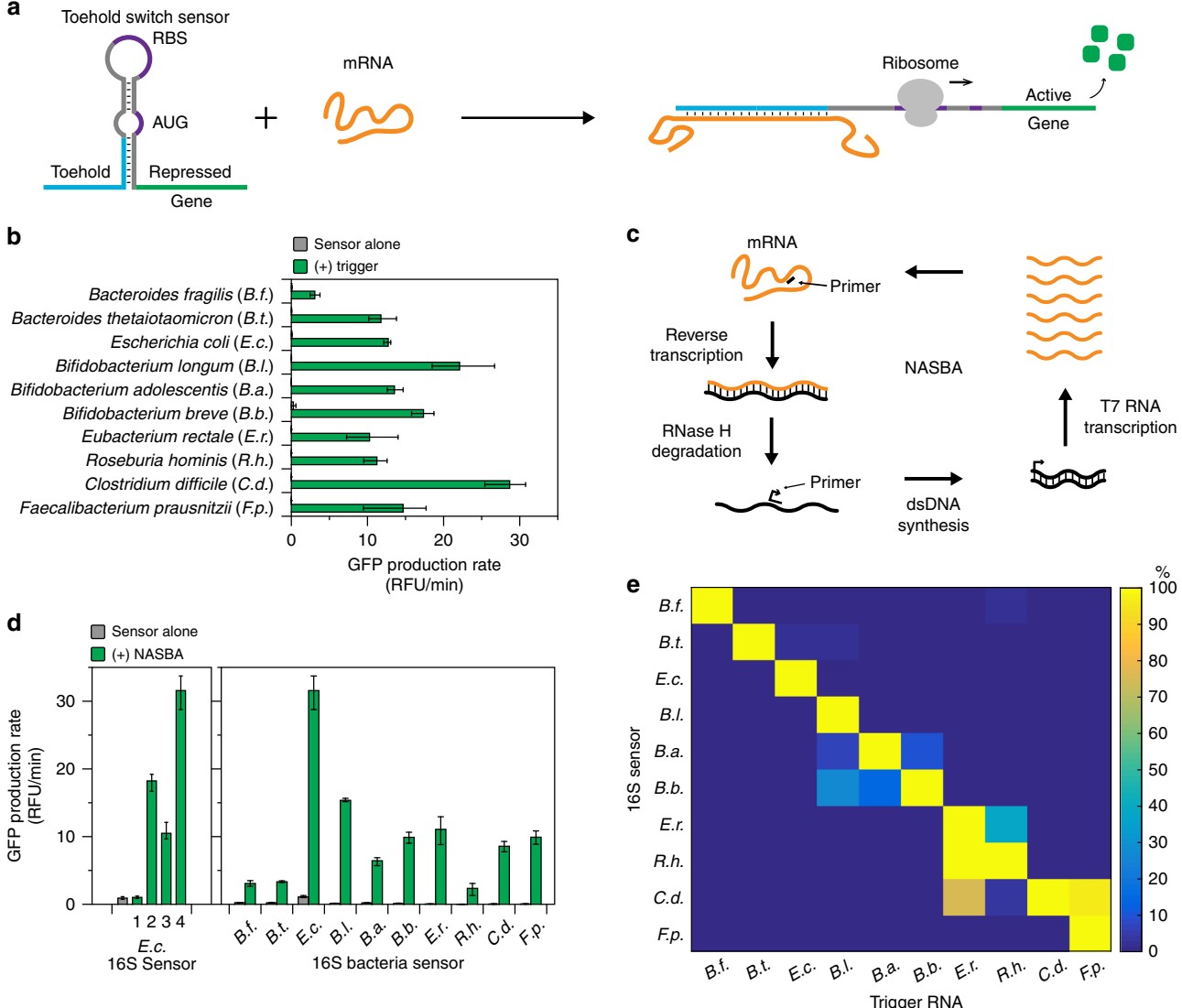

**Fig. 2** 16S rRNA sensors. **a** Schematic of toehold switch sensor function. **b** Best performing toehold switch sensors targeting the V3 hypervariable region of 16S rRNA for each species. Data represent mean GFP production rates from paper-based reactions with sensor alone and sensor plus 36-nucleotide trigger RNA (2 μM). Error bars represent high and low values from three technical replicates. **c** Schematic of NASBA-mediated RNA amplification. **d** Evaluation of NASBA primers. NASBA reactions were performed on 1 ng of total RNA for 90 min. Outputs from NASBA reactions were used to activate toehold switch sensors in paper-based reactions. Data represent mean values of three technical replicates. Error bars represent high and low values of the three replicates. **e** Orthogonality of 16S sensors. Each sensor was challenged with 2 μM of NASBA trigger RNAs from each species representing what would be amplified in a NASBA reaction. GFP production rates for an individual sensor were normalized to the production rate of the sensor plus its cognate trigger (100%). Data represent mean values of six replicates (two biological replicates × three technical replicates). Full data and s.d. are shown in Supplementary Figure 3

significant crosstalk detected between any of the sensors (Fig. 3c, Supplementary Fig. 6).

Next, we investigated the specificity of our NASBA primers by testing the output of NASBA reactions performed on three different total RNA samples: (1) total RNA isolated from an individual species; (2) a mixed sample comprised of total RNA from each of the 10 species; and (3) a mixed sample containing total RNA from all species except for the one corresponding to the NASBA primer set being tested. To keep the total concentration of a given sample constant, we supplemented samples (1) and (3) with yeast tRNA (Ambion), which is commonly used to increase the complexity of mRNA standards in RT-qPCR, because reverse transcription efficiencies change with the total amount of RNA in a reaction[31]. For example, each NASBA reaction was run on a total of 10 ng of RNA, where sample (1) contained 1 ng of total RNA plus 9 ng of yeast tRNA

and sample (2) included 1 ng of RNA from each of the 10 individual species. For each NASBA primer set, we observed equivalent activation of the toehold switch sensors by RNA amplified from samples (1) and (2). Additionally, the outputs from sample (3) were equivalent to the toehold switch sensor alone for each species indicating that there was no amplification of the test target in sample (3) (Fig. 3d). These results showed that the NASBA primers were highly specific within the tested set of 10 bacteria, which included closely related species.

**Toehold switch sensors quantify NASBA products.** Quantitation is essential for determining changes in bacterial and host gene expression and abundances of microbes. Therefore, we sought to determine if the toehold switch sensors could be used to quantify bacterial RNA in fecal samples. Previous work has shown that NASBA can be quantified using internal standards

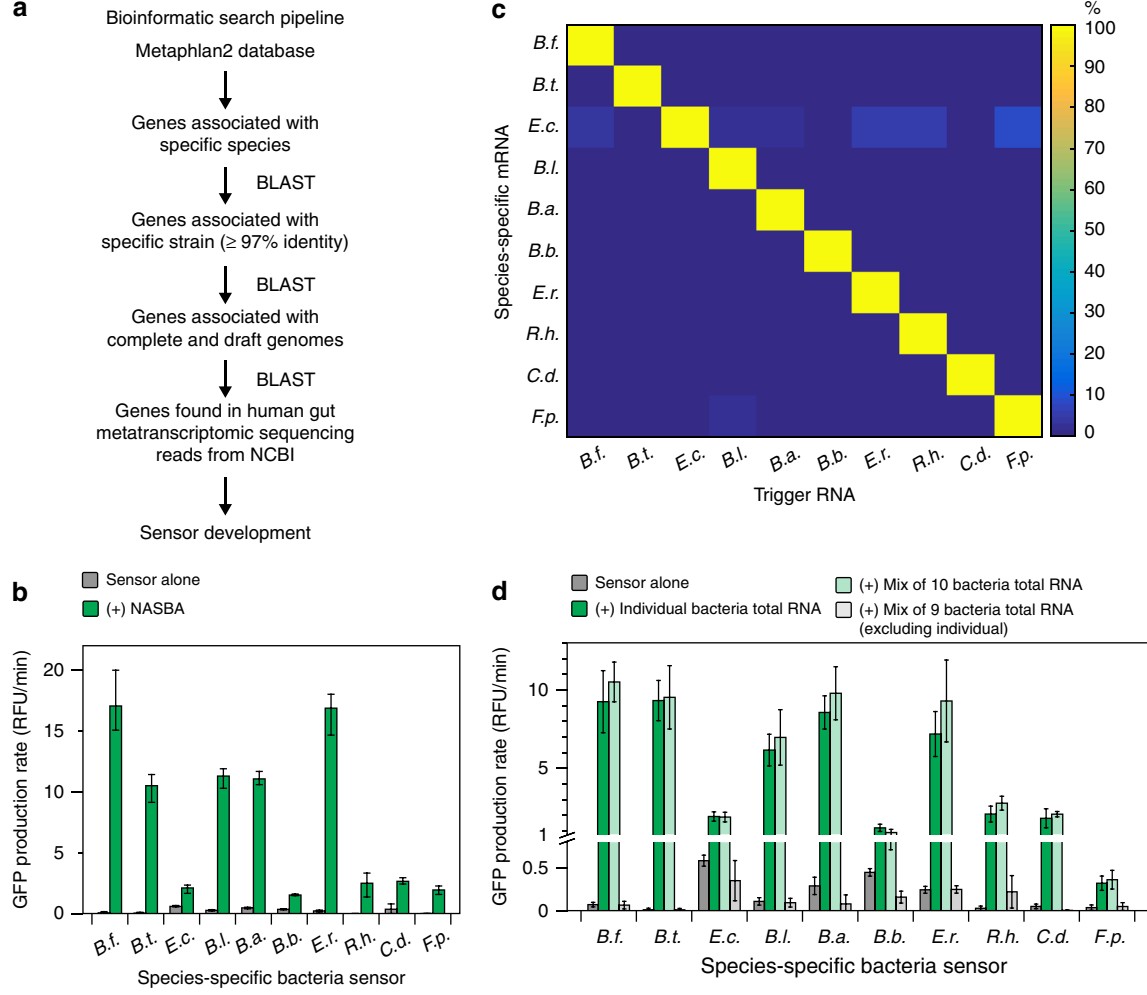

**Fig. 3** Species-specific mRNA sensors. **a** Bioinformatic pipeline for identifying species-specific mRNAs. **b** Best performing NASBA primers and species-specific mRNA sensors for each species. NASBA reactions were performed on 10 ng of total RNA for 90 min. Outputs from NASBA reactions were used to activate toehold switch sensors in paper-based reactions. Data represent mean values of three technical replicates. Error bars represent high and low values of the three replicates. **c** Orthogonality of species-specific sensors. Each sensor was challenged with 2 µM of trigger RNAs from each species representing what would be amplified in a NASBA reaction. GFP production rates for an individual sensor were normalized to the production rate of the sensor plus its cognate trigger (100%). Data represent mean values of six replicates (two biological replicates × three technical replicates). Full data and s.d. are shown in Supplementary Figure 6. **d** Orthogonality of NASBA primer sets. NASBA reactions were performed on 10 ng of total RNA for 90 min. Data represent mean ± s.d. of six replicates (two biological replicates (NASBA reactions) × three technical replicates (paper-based reactions))

and fluorescent hybridization probes to detect amplified RNA[32,33]. In a previous application of the paper-based diagnostic platform, we demonstrated that the toehold switch sensors exhibit a linear response to trigger RNA inputs in the low nanomolar to micromolar range[17]. A mathematical model of NASBA reactions suggested that femtomolar to picomolar concentrations of RNA could be amplified to within the toehold detectable linear range, and 10-fold concentration differences would be distinguishable if NASBA reactions were stopped prior to completion (Supplementary Fig. 7). Therefore, we sought to identify NASBA reaction conditions that would allow us to quantify a broad range of RNA concentrations using the toehold switch sensors.

We in vitro transcribed species-specific mRNAs and used them as standards for the NASBA reactions. We aimed to quantify standards from 3 fM to 30 picomolar (pM). To mimic the complexity of a total RNA sample, we diluted our standards into yeast tRNA (50 ng/µl). NASBA reactions with varied amplification times (30 min–3 h) were carried out on mRNA standards to determine the duration that allowed us to distinguish concentrations that differed by 10-fold (Supplementary Fig. 8).

Excessive amplification times or running amplification reactions to completion did not allow for differentiation between standards, and insufficient amplification times did not allow for detection of the lowest (3 fM) standard. Using the optimal amplification time for each mRNA, we assessed the run-to-run variability of NASBA and paper-based toehold reactions. We found that there is run-to-run variation in overall signal measured from the paper-based reactions, but the relative signal between standards remains the same between runs (Fig. 4a). Normalization to a single standard allowed us to define a calibration curve that eliminated the effect of run-to-run variability on RNA quantification (Fig. 4b). Calibration curves were determined for each of the 10 species (Supplementary Fig. 9). These allow for calculation of species-specific mRNA concentrations in an unknown sample by simply running a single concurrent standard.

To validate our calibration curves, we sought to compare RNA quantification from human stool samples using our paper-based platform and RT-qPCR. We first assessed our ability to detect target mRNA in a pool of total RNA extracted (RNeasy PowerMicrobiome kit, Qiagen) from commercial human stool

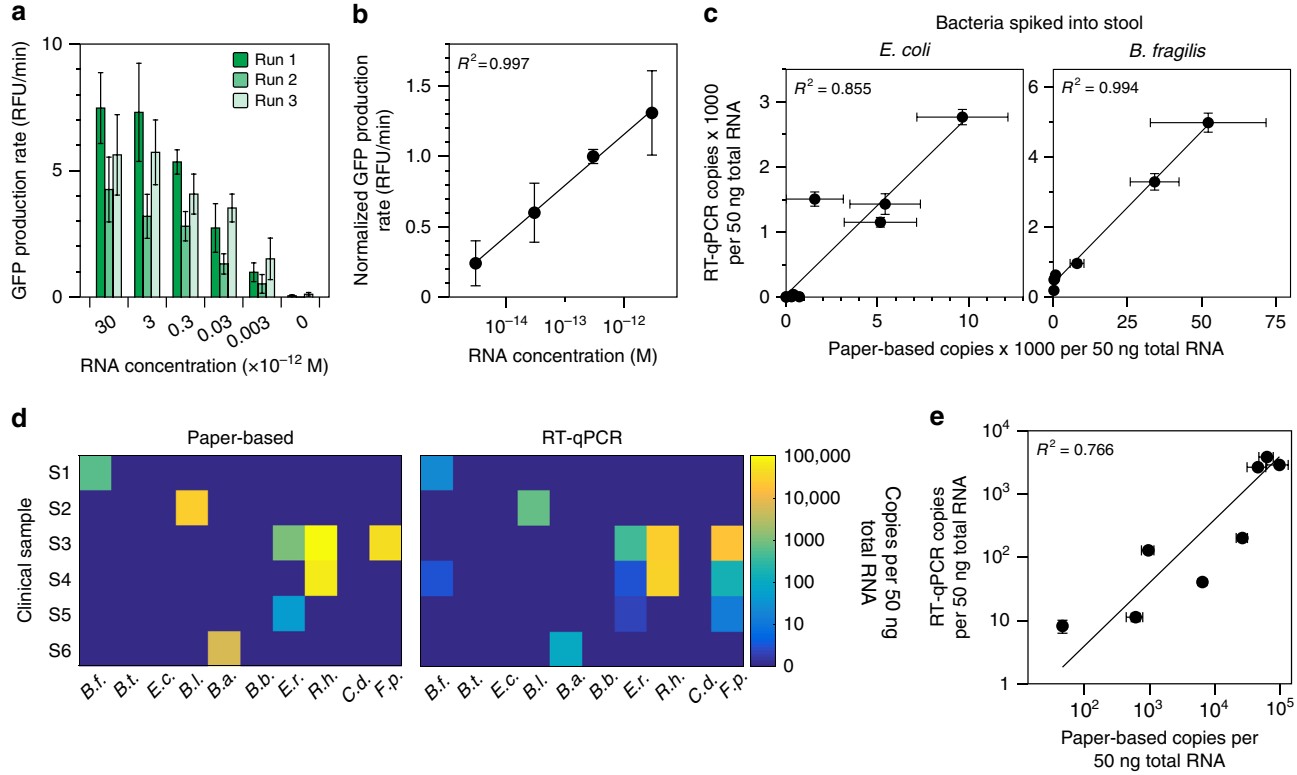

**Fig. 4** Quantification of NASBA-mediated amplification using toehold switch sensors. **a** Run-to-run variation in mRNA standards amplified by NASBA and measured by toehold sensors. mRNA standards for the *B. thetaiotaomicron* species-specific sensor were run in NASBA reactions for 30 min. Outputs from NASBA reactions were used to activate toehold switch sensors in paper-based reactions. **b** Calibration curve for the *B.t.* species-specific mRNA. Values from each standard in the individual runs in **a** were normalized to the 300 fM standard for that specific run and averaged across runs. **c** Quantifying species-specific mRNAs in stool. *E. coli* or *B. fragilis* cells were spiked into 150 mg of a commercial stool sample and processed for total RNA. Species-specific mRNAs were quantified using our paper-based platform and RT-qPCR. **d** Analysis of clinical stool samples. Six clinical stool samples were processed for total RNA and analyzed by our paper-based platform and RT-qPCR. Data and s.d. are shown in Supplementary Figure 11. **e** Correlation of clinical sample results. Non-zero paper-based concentrations from **d** were compared to RT-qPCR determined values. Data represent mean values. Paper-based error bars in **a**, **c**, and **e** represent s.d. from nine replicates (three biological replicates (NASBA reactions) × three technical replicates (paper-based reactions)). RT-qPCR error bars in **c** and **e** represent s.d. from six replicates (two biological replicates (RT reactions) × three technical replicates (qPCR reactions))

(Lee BioSolutions) and compared quantification of mRNA standards in this background to standards in a yeast tRNA background. We detected our species-specific mRNAs in stool RNA background, but the signal output for any given standard concentration was higher in total stool RNA than in the yeast tRNA background (Supplementary Fig. 10). Therefore, we experimentally corrected each of our calibration curves to account for this difference (Supplementary Fig. 9).

We then compared our quantification method to RT-qPCR. We spiked in between 50 μl and 1.5 ml of bacterial cells grown to mid-log phase to 150 mg of commercial human stool. These samples were processed for total RNA and quantified using our paper-based platform and RT-qPCR. We found good correlation between these methods with $R^2$ values of 0.855 and 0.994 for *E. coli* and *B. fragilis*, respectively (Fig. 4c).

Next, we tested the performance of our quantification method with clinically acquired stool samples (Fig. 4d). In the six clinical samples tested, we detected six of the bacteria in our panel. The concentrations of species-specific mRNAs determined using our platform showed good correlation with RT-qPCR, with an $R^2$ value of 0.766 (Fig. 4e). We had no false-positive results and seven false-negative results using RT-qPCR as the standard (Supplementary Fig. 11). Of the seven false-negative results, six contained less than three copies per 50 ng of total RNA (6 attomolar) quantified by RT-qPCR, a value below our limit of detection.

**Toehold switch sensors can detect human mRNA from stool.** Next, we sought to demonstrate that our platform could be used to detect mRNAs from human cells. We designed toehold switch sensors and NASBA primers to detect the mRNA of three biomarkers associated with inflammation (calprotectin, *CXCL5*, and *IL-8*) and oncostatin M (*OSM*), a cytokine that has recently been found to predict the efficacy of anti-tumor necrosis factor (TNF)-alpha therapies in IBD patients[20]. To validate our sensors, we performed NASBA and toehold reactions on 50 ng of total RNA from human peripheral leukocytes (Takara Bio 636592) and demonstrated that we could detect each of the four transcripts (Supplementary Fig. 12). We then developed calibration curves for each sensor (Supplementary Fig. 13) and tested the performance of our quantification method with clinically acquired stool samples from patients with IBD (Fig. 5a). We detected each of the four host transcripts in at least two of the clinical samples. Furthermore, the concentrations of human mRNA determined using our platform showed good correlation with RT-qPCR, with an $R^2$ value of 0.912 (Fig. 5b).

**RNA-based detection of *C. difficile* infection.** In a final validation of our platform, we sought to demonstrate the advantage of measuring RNA as opposed to DNA in certain clinical applications. CDI is one example where RNA-based detection may be especially useful. CDI causes significant patient morbidity and

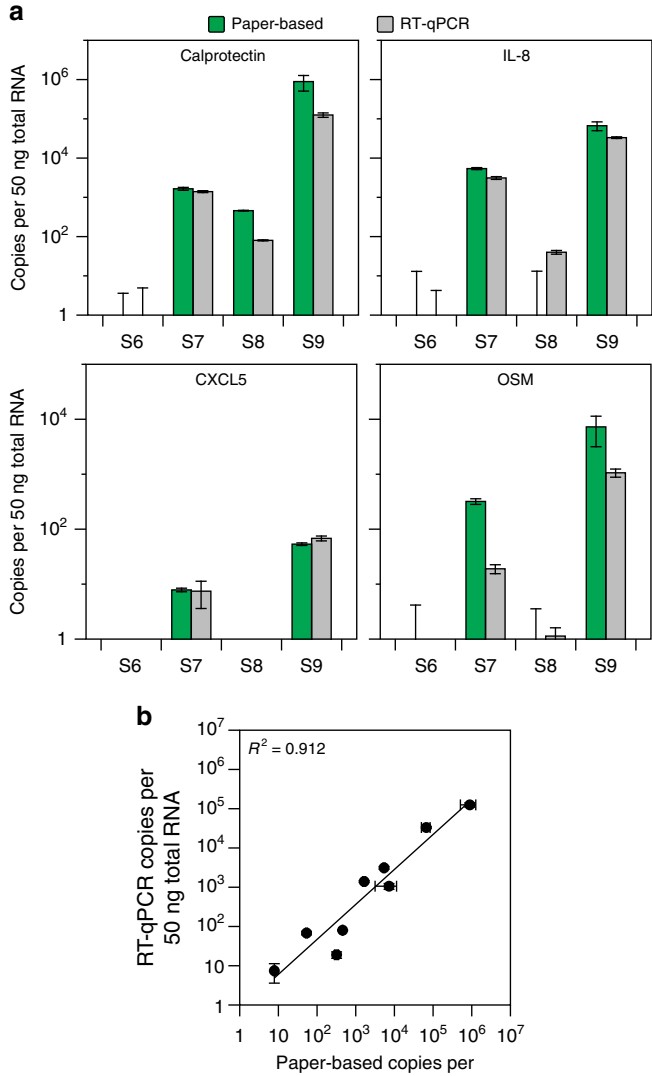

**Fig. 5** Detection of host biomarkers of inflammation. **a** Analysis of clinical stool samples. Four clinical stool samples were processed for total RNA and analyzed by our paper-based platform and RT-qPCR. Data represent mean values. Paper-based error bars represent s.d. from nine replicates (three biological replicates (NASBA reactions) × three technical replicates (paper-based reactions)). RT-qPCR error bars represent s.d. from six replicates (two biological replicates (RT reactions) × three technical replicates (qPCR reactions)). **b** Correlation of clinical sample results. Non-zero paper-based concentrations from **a** were compared to RT-qPCR determined values

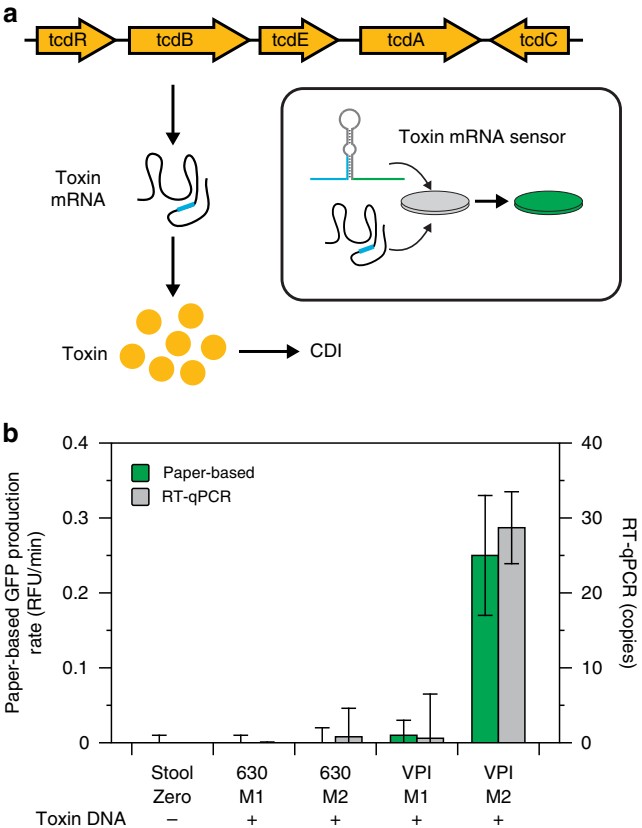

**Fig. 6** Paper-based detection of *C. difficile* infection. **a** Schematic of RNA-based CDI detection using a toehold switch sensor to detect toxin B mRNA. **b** Toxin B mRNA detection in stool samples. Two *C. difficile* strains (630 and VPI 10463) were grown in two different media (M1—TYG plus cysteine, M2—TY). Cells from each culture were spiked into 150 mg of a commercial stool sample and processed for total RNA. Toxin B mRNA was measured by our paper-based platform and RT-qPCR. Data represent mean values. Paper-based error bars represent s.d. from nine replicates (three biological replicates (NASBA reactions) × three technical replicates (paper-based reactions)). RT-qPCR error bars represent s.d. from six replicates (two biological replicates (RT reactions) × three technical replicates (qPCR reactions)). Toxin B DNA was confirmed in each sample using qPCR (Cq values shown in Supplementary Table 11)

mortality[34], and is responsible for nearly 2.4 million days of inpatient hospital stays at a yearly cost of over $6.4 billion in the United States[35]. CDI-associated diarrhea and intestinal inflammation are attributed to the direct effects of *C. difficile* toxins[36]. As such, current CDI diagnostic tests are focused on detecting the presence of toxigenic *C. difficile* bacteria or the toxin proteins in patient stool.

The traditional gold standard tests for detecting toxigenic *C. difficile* organisms (toxigenic culture assay) and *C. difficile* toxin (cell-culture cytotoxicity neutralization assay) are slow, labor-intensive, and technically challenging[37]. The diagnostics currently in wide-spread use, such as enzyme-linked immunoassays (EIA) for *C. difficile* toxins and DNA-based qPCR assays for *C. difficile* toxin genes, offer greatly improved performance characteristics but have their own limitations[21]. The EIA tests have high clinical

specificity, but reports of false-negatives and low sensitivity relative to toxigenic culture[21,37] have led to the development of DNA-based qPCR assays for *C. difficile* toxin genes. This method is extremely sensitive for the presence of toxigenic *C. difficile* bacteria; however, it cannot distinguish between patients that are carriers with symptoms due to another cause and those with active CDI[21]. These cases are especially challenging for clinicians, and there is a debate on which testing methodology yields the highest combination of sensitivity and specificity for clinically meaningful CDI[21]. New ultrasensitive assays to detect *C. difficile* toxins are in development, but they require highly specialized and expensive laboratory equipment and in some cases have a 60-h turnaround time[38]. Our paper-based platform has the potential to address these limitations by providing a rapid, easy-to-use method for the diagnosis of active CDI based on the detection of *C. difficile* toxin mRNA (Fig. 6a).

We designed a toehold switch sensor and NASBA primers to detect a conserved region of the *C. difficile* toxin B gene, which is essential for toxigenic effect and is the target of most commercial DNA-based qPCR assays for toxigenic *C. difficile*[39]. To validate

our sensor, we collected total RNA from monocultures of two different toxigenic *C. difficile* strains: 630, a low toxin producing strain, and VPI10463 (VPI), a high toxin producing strain[40]. We performed NASBA and toehold reactions on 25 ng of total RNA from each strain and demonstrated that we could detect toxin mRNA from both *C. difficile* strains (Supplementary Fig. 14). Next, we grew the two strains under conditions that suppress (mid-log phase in media 1: TYG plus cysteine) or induce (stationary phase in media 2: TY) toxin production to mimic situations where patients are carriers of toxigenic *C. difficile* that produce very low levels toxin and those with active CDI resulting from high toxin production, respectively. We then spiked the two strains grown in both conditions into commercial human stool and processed the samples for total RNA as described previously. Using our paper-based platform, we detected toxin mRNA only in the VPI strain grown in media 2 sample (Fig. 6b). Analysis of the samples using RT-qPCR indicated that there was toxin mRNA in the 630 media 2 and VPI media 1 samples, but at very low levels ($1 \pm 4$ and $1 \pm 6$ copies or 2 attomolar, respectively). Furthermore, all four samples were positive for toxin DNA (Supplementary Table 12). Our results therefore demonstrate a potential advantage of using toxin mRNA to diagnose CDI. All four samples would give a positive result in a DNA-based qPCR test. However, by detecting toxin mRNA using our paper-based platform, it may be possible to rapidly and readily distinguish between carriers of toxigenic *C. difficile* expressing low levels of toxins and those patients with active CDI expressing significantly higher levels of toxins.

## Discussion

Here we presented a synthetic biology platform for affordable, on-demand analysis of microbiome samples that can be employed in research, clinical, and low-resource settings. We demonstrated detection of species-specific mRNAs from 10 different bacteria that have been associated with a wide variety of disease processes. To track abundance of target RNAs, we devised a method to quantify mRNA using our toehold sensors and validated our method using RT-qPCR on clinical stool samples. To highlight the ability to probe both host and bacterial transcripts using a single platform, we validated sensors for clinically relevant human mRNAs using stool samples from IBD patients. We also demonstrated the potential advantage and clinical utility of detecting toxin mRNA in the case of CDI.

As part of this study, we developed a simple method that allows for the semi-quantitative determination of mRNA concentration from human stool samples using paper-based toehold switch sensors. By running a single standard alongside test samples and referencing a standard curve, we can determine the mRNA concentration within a sample and account for variation in reagent lots with clear separation of samples that differ in concentration by 10-fold (Fig. 4a, b). Our method is analogous to those used for NASBA-based quantification with an internal control spiked into each sample and a fluorescent hybridization probe for detection[32,33]. Furthermore, quantification of mRNAs in stool samples using our method correlates well with RT-qPCR (Fig. 4c–e, Fig. 5b). Notably, mRNA concentrations correlate with bacterial abundance (Supplementary Fig. 15), though this correlation may fluctuate with growth conditions and will likely vary depending on the specific target.

Our approach is easily adaptable to study any cellular process that results in differences in gene expression, including changes in specific biochemical pathways or cell metabolism. To illustrate the potential utility of assessing specific bacterial pathways, we selected the model of toxin production in CDI. To approximate the clinical scenarios of active CDI versus inactive colonization,

we demonstrated that we could distinguish between toxigenic *C. difficile* that expressed high amounts of toxin and no toxin (Fig. 6), which would otherwise be indistinguishable via standard DNA-based qPCR. Recent studies have shown that fecal mRNA levels of the inflammatory markers *CXCL5* and *IL-8* are highly correlated with clinical outcomes and perform with significantly better clinical sensitivity than other available tests for identifying CDI[41,42]. Because our method is equally capable of quantifying microbial and host RNAs and is readily multiplexed, a combined diagnostic testing for *C. difficile* toxin, *CXCL5*, and *IL-8* mRNA may provide improved sensitivity and specificity for detecting CDI, though further investigation using clinical samples is warranted to help address this important problem.

In addition to the potential utility of our platform in the clinical diagnosis of CDI, our ability to assess both host and microbial transcripts in parallel may also be useful in management and treatment selection for IBD. The interaction between the host and resident microbiome has been shown to affect many important biological processes in health and disease, including IBD[12]. Recent work has demonstrated that a microbial signature can be predictive of clinical remission after treatment with vedolizumab, an anti-integrin IBD medication[43]. For host transcripts, calprotectin is a well-characterized biomarker routinely used in clinical practice to assess gut mucosal inflammation[19]; *CXCL5* and *IL-8* are both elevated in intestinal biopsies from patients with IBD[44,45]; additionally, *OSM* levels in intestinal tissues have recently been strongly correlated with a lack of response to anti-TNF agents[20], a widely used class of medications to treat IBD. Although highly efficacious, roughly 30–40% of patients will not respond to the anti-TNF medication class, and there was previously no reliable way of predicting the likelihood of response. While the above study was based on intestinal biopsies, we demonstrated we could detect *OSM* mRNA from IBD patient stool samples. Although the low number of samples precludes any conclusions on clinical utility, our results are consistent with a connection between higher *OSM* levels and lack of responsiveness to anti-TNF treatment. For example, stool sample S6 with no detectable *OSM* mRNA was collected from a patient who had successfully responded to anti-TNF treatment. Furthermore, sample S7, which showed intermediate levels of *OSM* mRNA, was collected from a patient who had failed treatment with two different anti-TNF agents.

Our platform provides an easy to use, low-cost method for quantifying microbial and host RNAs from complex biological samples. Its flexibility allows for reactions to be freeze-dried for use outside of a laboratory setting. All reactions can also be run fresh, as they were done here, for researchers that do not have access to a lyophilizer. Specialized lab equipment is not required to develop our sensors or run the reactions. Since our toehold switch sensors can be used to regulate the production of any protein output, reactions may be monitored on a standard microtiter plate reader, if available, or an affordable, easy-to-build, portable electronic reader that quantifies change in absorbance from LacZ production[17]. To accommodate incubation temperatures required for NASBA (95 °C, 41 °C) and paper-based (37 °C) reactions, existing laboratory incubators or thermocyclers may be used, or affordable incubators can be built for use in low-resource settings. Altogether, the low-cost and portable nature of our platform makes it uniquely suited for use in resource-limited environments.

The major advantages of our platform over RT-qPCR are cost and the ability to analyze multiple RNA transcripts at once. Using our platform, we can quantify mRNAs in 3–5 h at a cost of approximately $16 per transcript using commercially available kits as reagents (accounting for triplicate reactions and mRNA standard). This can be reduced to under $2 per transcript by

using cell-free extracts prepared in-house, which are suitable for our platform (Supplementary Fig. 16), and individually sourcing NASBA reagents (Supplementary Fig. 17). The same analysis using RT-qPCR also takes 3–5 h, but costs approximately $140 per transcript. Our platform only requires a single mRNA standard for quantification while RT-qPCR generally requires a minimum of five standards[46]. Our limit of detection in total stool RNA ranges between 30 aM and 3 fM, depending on the specific toehold switch sensor. While this does not match the sensitivity of RT-qPCR (3 aM)[47], we believe there are applications where our current limits of detection are sufficient. Future optimization of toehold switch sensor design and NASBA reaction conditions may continue to improve this sensitivity.

In a comparison of our platform to next-generation sequencing we offer fast turn-around time, simple data analysis, and on-demand assessment of samples with no change in cost per sample. Average next-generation sequencing runs at core facilities range from $700–2000 per lane (Illumina), depending on machine and run type, and can take anywhere from 4 to 72 h[48] to complete. The sequencing cost per sample is typically reduced by running up to 96 samples per lane; however, this sample batching prevents on-demand analysis. Additionally, next-generation sequencing data sets require extensive computational power and training to process, analyze, and interpret. Our platform's data analysis can be performed quickly using a simple spreadsheet or automated program.

Our paper-based platform is one of several new synthetic biology platforms that can be used for nucleic acid detection. Recent advances using the CRISPR associated enzymes Cas12a and Cas13 along with recombinase polymerase amplification (RPA)[49] yielded sensitive detection of nucleic acids with the ability to discriminate between single nucleotide differences[50–52]. While detection of single nucleotide polymorphisms (SNPs) is important, for example in tracking the epidemiology of viruses, the ability of the toehold switch sensors to tolerate SNPs enables the use of a single sensor to detect multiple strains. Although RT-RPA can be used to amplify RNA, as with RT-qPCR it cannot specifically amplify RNA without thorough DNase treatment to remove genomic DNA. Since NASBA uses reverse transcription to create DNA with a T7 promoter to then transcribe that template into RNA, it is highly resistant to DNA contamination[53]. Our method and the CRISPR enzyme-based diagnostics, SHERLOCK[50,51] and DETECTR[52], could be complementary tools, the selection of which will depend on the sample type (DNA or RNA), and whether the detection of single nucleotide differences is desired.

Our method for detecting and quantifying RNA sequences could be applied to a broad range of studies including samples from other human anatomical sites, and our approach is easily adaptable to a wide range of biological targets, including viruses, fungi, and eukaryotic nucleic acids from either stool or tissue samples. Furthermore, with continued optimization of sample processing, our method could be adapted for point-of-care use. Such a diagnostic platform could have many applications, including pre-screening enrollees in the field for prospective trials of therapeutic manipulations of the microbiome, at-home monitoring of research participants, and eventually for tracking changes in patient disease activity. Our easy-to-use synthetic biology platform has the potential to meet both research and clinical point-of-care needs.

## Methods

**Toehold sensor design and cloning.** Toehold switch sensors were designed with NUPACK[24] using the series B toehold switch design from Pardee et al.[17] The script can be found in Supplementary Note 1. Toehold switch sensor designs were checked for premature stop codons and cloned into plasmids with the GFPmut3b

gene using PCR amplification and blunt-end ligation. Linear toehold switch templates were generated by amplifying from these plasmids by PCR and then purified using a MinElute PCR Purification kit (Qiagen, 28004), according to manufacturer's protocol. Sequences for all toehold switch sensors can be found in Supplementary Tables 2–3.

**Trigger RNA and mRNA standard synthesis.** DNA encoding trigger RNAs or mRNA standard sequences were ordered from Integrated DNA Technologies and amplified by PCR to create a linear template with a T7 promoter. RNA was transcribed from the DNA templates using a HiScribe T7 High Yield RNA Synthesis Kit, according to the manufacturer's protocol (New England Biolabs, E2040). RNA was then purified using a Zymo RNA Clean and Concentrator kit (R1018), according to the manufacturer's protocol. Following purification, DNA template was degraded by DNase digestion using the TURBO DNA-free DNase kit (ThermoFisher, AM1907) for 1 h according to the manufacturer's protocol.

**Paper-based, cell-free reactions.** Cell-free reactions were performed using the PURExpress In Vitro Protein Synthesis Kit (New England Biolabs, E6800L). The cell-free reactions consisted of NEB Solution A (40%), NEB Solution B (30%), RNase inhibitor (0.5%; Roche, 3335402001), linear DNA constructs encoding toehold switch sensors (1.875 nM), and trigger RNA for a total of 5.5 µl. Paper disks (Whatman, 1442-042 blocked overnight in 5% BSA) were punched out using a 2 mm biopsy (Integra, 33-31-P/25) and placed in a 384-well plate (Corning 3544). 1.4 µl of the cell-free reaction mixture was applied to paper disks in triplicate. GFP expression (485 nm excitation, 520 nm emission) was monitored on a plate reader (Molecular Devices SpectraMax M5) every 5 min for 2 h at 37 °C.

**Initial sensor screen.** Sensor candidate designs from NUPACK were tested in paper-based reactions containing 1.875 nM of linear sensor DNA and 2 µM trigger RNA (36 nucleotides). GFP production rates were calculated (see Data analysis and RNA quantification) for reactions with sensor alone and sensor plus trigger. To select the best sensor, an activation ratio was calculated for each sensor candidate by dividing the sensor plus trigger production rate by the sensor alone production rate. Sensors were chosen based on the highest activation ratio and lowest sensor alone production rate. A minimum activation ratio of 5-fold is necessary to achieve desired sensitivity.

**NASBA.** Initial denaturation of total RNA consisted of a 2-min incubation at 95 °C followed by a 10-min incubation at 41 °C of 1.0 µl sample input, 1.675 µl reaction buffer (Life Sciences Advanced Technologies, NECB-24), 0.825 µL nucleotide mix (Life Sciences Advanced Technologies, NECN-24), 0.2 µl of 6.25 µM primers, 0.03 µl water, and 0.025 µl of RNase inhibitor (Roche) per 3.75 µl reaction. Afterwards, 1.25 µl of enzyme mix (Life Sciences NEC-1-24) was added to each reaction and the resulting 5.0 µl NASBA reactions were incubated for 30–180 min at 41 °C. Then 1.0 µl of NASBA product was added to the cell-free reaction mixture for a total of 5.5 µl.

Final concentrations of buffer components in each NASBA reaction: 13.2 mM MgCl$_2$ (VWR 97062-848), 75 mM KCl (VWR BDH7296-0), 10 mM DTT (Sigma GE17-1318-01), 40 mM Tris-HCl pH 8.5 (VWR RLMB-005), 15% DMSO, 2 mM each ATP, UTP, and CTP, 1.5 mM GTP, 0.5 mM ITP, 1 mM each dNTP (New England Biolabs, N0447L), 0.25 µM each primer. Enzyme mix: 5 U/ml RNaseH (New England Biolabs M0297L), 1000 U/ml reverse transcriptase (New England Biolabs, M0368L), 2500 U/ml T7 RNA polymerase (New England Biolabs, M0251L), 43.75 mg/ml BSA. Initial denaturation of sample was performed as above, after which 1.25 µl enzyme mix was added to each reaction.

**Data analysis and RNA quantification.** Paper-based reactions were analyzed by calculating GFP production rates for each reaction condition. GFP production rates were calculated by first subtracting the average background fluorescence measured from triplicate paper-based reactions that did not contain sensor DNA or trigger RNA. Then, the minimum value of each individual reaction was adjusted to zero by subtracting the average of its first three time points (0, 5, and 10 min) from each time point. The zero-adjusted data were then fit to the equation: $RFU(zero\ adjusted) = \frac{a}{e^{-bt} + c}$. To compare data from different samples, the slope of the fitted equation was taken at $t = 50$ min, resulting in values of RFU/min. The GFP production rates were then averaged over the replicates for each reaction condition.

In quantification experiments, the GFP production rate for each sample was normalized to the GFP production rate for a single mRNA standard (for standard concentrations see Supplementary Fig. 9). The normalized GFP production rate for reactions with sensor alone was then subtracted from each sample. RNA concentration was determined using the equation: Normalized GFP production = A*ln(concentration) + B. Values for A and B for each sensor can be found in Supplementary Fig. 9.

**Bacterial culturing and RNA processing.** All anaerobic bacteria were grown in an anaerobic chamber at 37 °C. Bifidobacterium adolescentis (ATCC 15703), Bifidobacterium breve (ATCC15700), Bifidobacterium longum subsp longum (ATCC

15707), Bacteroides fragilis (ATCC 25285), Bacteroides thetaiotaomicron (ATCC 29148), Clostridium difficile (ATCC BAA-1382), and Eubacterium rectale (ATCC 33656) were obtained from ATCC. Faecalibacterium prausnitzii A2–165 (DSM 17677) and Roseburia hominis (DSM 16839) were obtained from DSMZ. Freeze-dried samples were rehydrated with their respective growth mediums and grown for 24–48 h in liquid culture on a shaker at 200 rpm. For experiments testing RNA isolated from pure cultures, 12 ml of bacterial culture was diluted 1:2 into RNA-Protect before removing from the anaerobic chamber for RNA extractions.

The cultures were lysed at room temperature using 200 μl of 15 mg/ml of lysozyme in TE buffer and 20 μl of proteinase K (Qiagen). RNA was then extracted using the RNeasy Mini kit (Qiagen 74104), according to the manufacturer's instructions. RNA samples were then DNase digested using TURBO DNA-free DNase kit (ThermoFisher, AM1907) for one hour.

E. coli (MG1655) was grown in Luria-Bertan (LB) medium (Difco). B. adolescentis was grown in Bifidobacterium medium (prepared according to DSMZ 58: Bifidobacterium medium). B. breve, B. fragilis, and B. longum subsp longum were grown in brain heart infusion-supplemented (BHIS) medium (prepared according to ATCC medium: 1293). B. thetaiotaomicron, E. rectale, and R. hominis were grown in cooked meat medium (CMM) purchased from Hardy Diagnostics. F. prausnitzii was grown in CMM with an additional 1% glucose. C. difficile was grown in BHIS for the species-specific RNA testing, and grown in either TY medium (3% tryptone, 2% yeast extract, and 0.1% sodium thiogrlycolate), or TY medium with 2% glucose and 10 mM cysteine for toxin RNA testing.

**RNA purification from stool samples.** Commercial stool specimens were purchased from Lee Biosolutions and provided as frozen specimens. Clinical stool samples were provided by Dr. Ashwin Ananthakrishnan as anonymized specimens from the Prospective Registry in IBD Study at Massachusetts General Hospital. Approval was provided by the Partners Healthcare Human Subjects Research Committee. Informed consent was obtained from all subjects. Both commercial and clinical stool samples were stored at −80 °C and processed using the RNeasy Powermicrobiome Kit (MoBio, now Qiagen, 26000), which was selected for its ability to isolate high quality RNA from stool[54]. Each frozen stool was homogenized using a mortar and pestle cooled with liquid nitrogen[55], and 150 mg of each sample was loaded into each glass bead tube. Mechanical lysis was performed using a MoBio vortex adapter and a Vortex Genie 2 (Scientific Industries Inc) at maximum speed for 10 min. The manufacturer's protocol was followed for RNA extraction with optional on-column DNase digestion included. Resulting RNA samples were then further DNase digested using TURBO DNA-free DNase kit (ThermoFisher, AM1907) for one hour.

**Bacterial spike-in experiments.** E. coli and B. fragilis were grown to mid-log phase and spiked into a commercial stool sample (Lee Biosolutions) before RNA extraction. Bacteria cultures ranging from 10 μl to 1.5 ml were spun down before being re-suspended in PM1 buffer and added to 150 mg of stool. C. difficile was grown to stationary phase in TY medium and TY medium supplemented with 2% glucose and 10 mM cysteine. Two ml of stationary C. difficile culture was spun down and re-suspended in PM1 buffer and added to 150 mg of stool. All samples were processed with the RNeasy PowerMicrobiome kit, according to the manufacturer's instructions, with an extended 30-min lysis step for C. difficile spike-ins. RNA samples were then split into two samples, one that was DNase digested with the TURBO DNA-free DNase kit (ThermoFisher, AM1907) for one hour and one that did not receive DNase treatment so that it could be used for DNA based qPCR.

**Computational pipeline for species-specific RNA sequences.** Our computational pipeline employs components from previously developed phylogenetic assignment tools, including Metaphlan and Metaphlan2[29]. These programs use multiple bioinformatics approaches to reduce each bacterial species to a "bag of genes" and identify the set of genes or gene parts that is specifically associated with a target species or clade and not associated with any others. We extracted the Metaphlan2 markers for a given target species and used BLAST[30] alignments against available genomes for our target species to ensure that the markers were present. We then assessed these preliminary markers for expression in the human fecal microbiome by using BLAST alignments against a human stool transcriptome database that we created from repositories of publically available adult human stool meta-transcriptome sequencing reads. Keeping only the markers that are expressed in human stool, we again tested for specificity by performing BLAST alignments against a pan-bacterial database that we created from all publically available reference and draft bacterial genomes. We selected the most specific markers with the highest expression in human stool and created toehold switch sensors to target these RNA sequences. In the case of C. difficile, expression was extremely low for all Metaphlan2 markers from the standard human stool transcriptome database. This was not unexpected since this species is reported to be very lowly abundant in normal healthy populations. To develop sensors for this species, we instead screened for expression using transcriptomic data from C. difficile cultures in various conditions available in public repositories.

**RT-qPCR validation.** RNA from stool samples and in vitro transcription was extracted, purified, and DNase digested as described above. In vitro transcribed

RNA diluted in 150 ng/μl yeast tRNA (ThermoFisher, AM7119) was used to generate standards for absolute quantitation based on calculations designed to incorporate 1 to $10^7$ RNA copies per reverse transcription reaction. cDNA synthesis from stool samples was performed with 300 ng input RNA per reaction using Superscript III (ThermoFisher, 18080-400), according to the manufacturer's protocol, using gene-specific primers (reverse qPCR primers as indicated in Supplementary Table 8) at a final concentration of 2 μM and total volume of 20 μl. Quantitative PCR reactions were prepared in triplicate using 2 μl of the RT reactions, LightCycler 480 Probes Master Mix (Roche, 04707494001), primers (final concentration 5 μM) and hydrolysis probe (final concentration 1 μM) in a reaction volume of 20 μl. The qPCR reactions were performed on a Lightcycler 480 96-well machine using the following program: (i) 95 °C for 10 min, (ii) 95 °C for 10 s, (iii) 48–60 °C for 50 s depending on primer Tm, (iv) 72 °C for 1 s for fluorescence measurement, (v) go to step ii and repeat 44 cycles, and (vi) 40 °C for 10 s. Absolute quantitation was performed using LightCycler 96 software version 1.1.0.1320 (Roche). When there were discordant results between triplicate amplification repeats, non-amplified reaction Cqs were set to 45 (equal to the total number of amplification cycles) prior to incorporation in copy number calculations. Dilution series reactions were performed on RNA extracted from several stool samples to demonstrate the absence of inhibition for the RT and qPCR reactions. Primers and probes used for RT-qPCR analyses are listed in Supplementary Table 8. Hydrolysis probes had a 5' 6-FAM dye, internal ZEN quencher after the 9th base, and 3' Iowa Black quencher (Integrated DNA Technologies). For Oncostatin M, we used the TaqMan RNA-to-Ct 1-Step Kit (ThermoFisher 4392653) with the commercially available probe set Hs00968300_g1 (ThermoFisher 4331182).

**In-house cell-free extract preparation.** Cell extract was prepared as described by Kwon and Jewett[56]. E. coli BL21(DE3)ΔlacZ (gift of Takane Katayama) were grown in 400 ml of LB at 37 °C at 250 rpm. Cells were harvested in mid-exponential growth phase (OD$_{600}$ ~ 0.6), and cell pellets were washed three times with ice cold Buffer A containing 10 mM Tris-Acetate pH 8.2, 14 mM magnesium acetate, 60 mM potassium glutamate, and 2 mM DTT, and flash frozen and stored at −80 °C. Cell pellets were thawed and resuspended in 1 ml of Buffer A per 1 g of wet cells and sonicated in an ice-water bath. Total sonication energy to lyse cells was determined using the sonication energy equation for BL21 Star$^{TM}$ (DE3), [Energy] = [Volume (μL)] − 33.6] × 1.8$^{-1}$. A Q125 Sonicator (Qsonica) with 3.174 mm diameter probe at a frequency of 20 kHz was used for sonication. A 50% amplitude in 10 s on/off intervals was applied until the required input energy was met. Lysate was then centrifuged at 12,000 rcf for 10 min at 4 °C, and the supernatant was incubated at 37 °C at 300 rpm for 1 h. The supernatant was centrifuged again at 12,000 rcf for 10 min at 4 °C, and flash frozen and stored at −80 °C until use. Using a previously published cell-free reaction protocol[57], reaction mixtures were composed of 26.6 % (v/v) of in-house lysate, 1.5 mM each amino acid except leucine (1.25 mM), 1 mM DTT, 50 mM HEPES (pH 8.0), 1.5 mM ATP and GTP, 0.9 mM CTP and UTP, 0.2 mg/mL tRNA, 0.26 mM CoA, 0.33 mM NAD, 0.75 mM cAMP, 0.068 mM folinic acid, 1 mM spermidine, 30 mM 3-PGA, 2% PEG-8000, 0.5 % (v/v) Protector RNase Inhibitor (Roche), 2 nM LacZ sensor plasmid DNA, 2 uM RNA trigger, and 0.6 mg/mL chlorophenol red-ß-D-galactopyranoside (CPRG, Sigma Aldrich, 59767) for lacZ sensor. Optimal potassium glutamate (40–140 mM) and magnesium glutamate (2–8 mM) concentrations were determined for lacZ reporter product. Reactions were first assembled on ice without CPRG, incubated at 37 °C for 30 min, chilled on ice for 5 min, and then CPRG was added to reaction. 1.4 μl of reaction mixture was then applied to pre-blocked 5% BSA 2 mm paper disks, placed in a black, clear bottom 384-well plate (Corning, 3544) and incubated at 37 °C for 1.5 h for the detection of lacZ expression.

**Code availability.** All code used in this work is available in Supplementary Note 1 and 2.

**Data availability.** All toehold switch sensors from this work have been deposited at AddGene. AddGene #110696-110717, 111907-111909. All other data supporting the findings of this study are available within the article and its Supplementary Information files, or are available from the authors upon request.

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

## Acknowledgements

This work was supported by MIT's Center for Microbiome Informatics and Therapeutics, the Paul G. Allen Frontiers Group, and the Wyss Institute. X.T. is supported in part by grants from the National Institutes of Health T32 DK007191 and a Wyss Institute Clinical Fellowship. A.J.D. is supported by the National Science Foundation Graduate Research Fellowship Program. A.A. is supported in part by grants from the National Institutes of Health (K23 DK097142, R03 DK112909) and the Crohn's and Colitis Foundation. The authors would like to thank Liz Andrews, Will Tan, Heather Wilson, and Eric Rosenberg for their help with clinical stool samples.

## Author contributions

M.K.T, X.T. and A.J.D designed experiments, performed experiments, analyzed data, and wrote the manuscript. D.B. performed experiments and edited the manuscript. Y.F. wrote code for identifying species-specific mRNA sequences. R.T.A. and N.D. performed experiments. A.A. provided clinical samples. J.J.C. directed overall research and edited the manuscript.

## Additional information

**Competing interests:** J.J.C. is an author on a patent application for the paper-based synthetic gene networks US20160312312A1 and a patent for the RNA toehold switch sensors US9550987B2. The remaining authors declare no competing interests.

