## [Peer Review File · Nature Communications]

Reviewers' comments:

Reviewer #1 (Remarks to the Author):

Summary:

This work "On-demand analysis of the gut microbiome using paper-based RNA sensors" builds upon previous work from the Collins lab in detecting Zika viral RNAs on paper-based sensors by applying it to analysis of the gut microbiome. Using a similar strategy involving NASBA and RNA toehold-mediated strand displacement reactions, the authors designed a series of toehold sensors against 10 common gut bacteria species, first targeting their 16S rRNAs, and when these showed a high degree of crosstalk, a new set computationally predicted to demonstrate orthogonality. The final pool of sensors and the corresponding NASBA primers designed to amplify the cognate RNAs show good ON/OFF induction and specificity. Next, the authors aim to show that their assay is quantitative. By including an internal RNA standard in each reaction and operating in a regime where the switch sensors show a linear response to RNA input, they were able to develop calibration curves for each species trigger RNA-sensor combination. These curves were validated by correlation against RT-qPCR, the gold standard for RNA quantification. Further, in clinical stool samples, the quantification showed agreement with the RT-qPCR results, including qualitative detection of particular species across six different samples, with false negatives mostly occurring only due to the limit of detection. Finally, the authors show that their platform can be used to detect *Clostridium difficile* infection by discriminating between toxigenic CDI infections and non-toxigenic infections, distinguished by the biosynthesis of the toxin B gene. Since the assay directly reports on mRNA levels rather than DNA levels, this should be an improvement over state-of-the-art assays that often present false positives in patients who carry the gene but do not have an active CDI infection.

Conclusions:

The authors demonstrate an effective use of paper-based systems for gut bacteria detection that could almost certainly be expanded limitlessly to detect other new species. The sensors show suitable detection thresholds, almost on par with state-of-the-art qPCR methods when isothermal amplification is used, and high orthogonality. From a practical perspective, the assay is cheap, easy-to-use, and rapid (with readout in just a few hours). However, the claim that the assay is quantitative is muddier (see below). In addition, the novelty, given past work seems in paper sensors from the group, seems somewhat low. While the paper has an exciting take home message, some major issues would need to be addressed if it were to be published in Nature Communications.

Major Issues:

- The authors claim that in a previous paper, they demonstrate that "the toehold switch sensors exhibit a linear response to trigger RNA inputs in the low nanomolar to micromolar range." Yet in Figure 4b, following NASBA RNA amplification, the ON state varies with the logarithm of RNA concentration. An isothermal amplification strategy, if run to completion, would seem to make the final RNA concentrations closer together regardless of the starting seed concentration, which begs the question of how this experimental dependence originates. Can the authors make or justify a model that suggests that GFP production vs.

starting RNA concentration before isothermal amplification should vary with a logarithmic dependence?

- The correlation curves developed in Figure 4b and Supplementary Figure 7 are plotted across very different concentration ranges and with a different number of points. Considering that many of these correlations appear poor, with large error bars, (particularly *B. longum*, which fits two parameters on three points), it does not seem fair to call this technique quantitative. This is particularly the case for Figures 4c and e, where the agreement between RT-qPCR and the predicted paper-based diagnostic diverge by factors of 3, 12, and 30, solely based upon the bacterial species and sample tested (spiked-in bacteria vs. clinical sample). If this technique cannot reliably predict the gold standard RT-qPCR reading within a factor of 5, then it cannot be called quantitative in my opinion. The positive results in Fig. 4d suggest that the word "semi-quantitative" is a fine alternative. Alternatively, if the authors can justify why RT-qPCR is underestimating the RNA concentration in the sample, this would be adequate.

Minor Issues:

- The authors claim that the in-house prepared extract behaves "suitabl[y] for our platform", yet the results in Fig. S12 suggest otherwise. The use of lacZ rather than GFP as a reporter is explained as "flexibility of the platform"—can the authors show results demonstrating that GFP is a sufficiently good reporter, considering the very low fold activation in this (enzyme) system compared to the complementary results in Fig. 3b? Otherwise, the authors should perhaps acknowledge that it does not work in the body of the manuscript.
- Since the publication of SHERLOCKv2, please adapt the wording of the introduction's claim that "We demonstrated the utility of our platform in detecting....but we were not able to quantify their concentrations."
- The authors "mapped the primer locations to chemical structure probing data for *E. coli* 30S ribosomal subunits". Please include these data in the text.

Reviewer #2 (Remarks to the Author):

Takahashi et. al. describe application of their toehold switch sensors and isothermal RNA amplification technique (NASBA) to detect bacterial transcripts in their manuscript, "On-demand analysis of the gut microbiome using paper-based RNA sensors". Two potential clinical diagnostic scenarios are described; the quantitative detection of marker RNAs from a panel of fecal bacteria (a surrogate for more extensive metagenomic studies) and the quantitative detection of *C. difficile* toxin mRNA in fecal samples.

The current manuscript builds upon the laboratory's prior exciting publications on toehold switch sensor and NASBA technology. Therefore the submission is not a proof-of-principle description of major new methodology per se. Yet, the elegant technology has been applied here to growing or well developed diagnostic targets, and as such hold great promise. Quantitative detection of *C. difficile* tcd gene abundance is likely to be among the most important applications of the RNA-targeted quantitation methodology. It is notable that the

assays could also likely be adapted to detection of host transcripts in clinical material. Such dual host-microbial assays would be novel and likely to have enhanced utility over microbe-targeted detection alone. For example, detection of host inflammatory transcripts would likely help distinguish *C. difficile* colonization from active *C. difficile*-associated colitis. Further, simultaneous detection of host and bacterial transcripts applied to clinical metagenomics will likely have major utility in a number of disease states. The NASBA-toehold switch sensor methodology may be ideal in those capacities, given the low cost of the assay reagents.

In addition to their tour de force application of RNA biology to clinical diagnostics, the manuscript describes several optimization steps which dramatically improve assay performance. In particular, the manuscript details optimization of the selectivity of various NASBA and sensor targets. Internal controls were developed and described which account for variability in signal strength and for background signal. Here, there are remaining concerns regarding assays performance. The authors describe significant continued run-to-run variability in signal response, as well as high background when run in clinical materials, and decreased sensitivity relative to qRT-PCR.

Taken together, the data presented here suggest that the technology holds great promise but is not yet sufficiently robust to be applied as diagnostic assay for either of the two described scenarios. As improvements in assay performance continue to occur, these assays will indeed likely have important clinical utility, and there are likely to be even broader clinical applications of NASBA-toehold switch sensor technology yet to be described.

Reviewer #3 (Remarks to the Author):

The work by Takahashi et al describes the development of a paper-based RNA sensor system to quantify microbiota samples in a cheap and direct way. The authors leveraged a series of innovations previously published by their group that combined toe-hold sensors, cell-free reactions, and paper-based deployment and extends the method to measure individual microbiota species from synthetic and natural communities. The authors further develop orthogonal sensors that can detect each of 10 distinct target species without cross-talk signals. Finally, a demonstration of the system for use in detection of *C. difficile* levels is presented.

Overall, the manuscript is clear, succinct, and well written and the data supports the claims of the work. The methodology utilized in the study is not particularly novel as it has been previously published. Nonetheless, the application of the strategy to quantify gut microbiota samples is novel. While the reviewer appreciates the interesting demonstration of the system for detecting 10 model gut bacteria, the practical use of such a detection system for research or clinical application is somewhat of a stretch, at least in the context of quantifying commensal bacteria. In comparison to RT-PCR methods, the current approach (including use of NASBA for better signal detection) seems to take a little bit longer and requires running standards for each reaction run. The cost saving is marginal to RT-PCR methods. RT-PCR requires a dedicated instrument although the GFP-based toehold sensor

still requires a quantitative UV-vis spectrophotometer for measurement. Both RT-PCR and toehold sensors are specific to the target species and thus individual primers and reaction conditions need to be developed. This of course is in contrast to Next-Gen Sequencing approaches which is more involved, but does provide much greater analytical power and throughput. All of those things said, the authors propose a compelling application of their method to detect a clinical pathogen, which has fast turnaround, high sensitivity, and can distinguish between high and low toxin expressing variants. What can be achieved in this paper for clinical pathogens goes beyond current DNA-based methods. As such, it would seem that expansion of this study to other clinical pathogens such as CRE, VRE, etc. would further strengthen this work in a revised form. Further demonstrating and comparing the results with RNA toehold on clinical samples to detect different pathogens (and not just in a spike-in experiment of the clinical sample as demonstrated) and comparing with current RT-PCR based methods would show direct translational application to boost the study's impact. Overall, the work is promising and systematically presents a protocol to detect microbiota from complex samples using a single RNA-based toehold sensor system that is easy to deploy in resource limited environments.

Minor points:

- 1) The authors should detail how the original toehold designs were generated for each of the species (i.e. Suppl Fig 1) and which ones were finally chosen for the subsequent studies (Fig 2b).
- 2) Some better quantitative analysis of the signal to noise (e.g. real signal vs background activity) could be applied or at least detailed. Currently, the rationale for choosing what a good design is seems to be somewhat qualitative. Quantitative treatment could help guide future users to generate suitable and high-performing toehold designs.

Point-by-Point Reviewer Response

Reviewer #1

Remarks to the Author:

Summary:

This work "On-demand analysis of the gut microbiome using paper-based RNA sensors" builds upon previous work from the Collins lab in detecting Zika viral RNAs on paper-based sensors by applying it to analysis of the gut microbiome. Using a similar strategy involving NASBA and RNA toehold-mediated strand displacement reactions, the authors designed a series of toehold sensors against 10 common gut bacteria species, first targeting their 16S rRNAs, and when these showed a high degree of crosstalk, a new set computationally predicted to demonstrate orthogonality. The final pool of sensors and the corresponding NASBA primers designed to amplify the cognate RNAs show good ON/OFF induction and specificity. Next, the authors aim to show that their assay is quantitative. By including an internal RNA standard in each reaction and operating in a regime where the switch sensors show a linear response to RNA input, they were able to develop calibration curves for each species trigger RNA-sensor combination. These curves were validated by correlation against RT-qPCR, the gold standard for RNA quantification. Further, in clinical stool samples, the quantification showed agreement with the RT-qPCR results, including qualitative detection of particular species across six different samples, with false negatives mostly occurring only due to the limit of detection. Finally, the authors show that their platform can be used to detect Clostridium difficile infection by discriminating between toxigenic CDI infections and non-toxigenic infections, distinguished by the biosynthesis of the toxin B gene. Since the assay directly reports on mRNA levels rather than DNA levels, this should be an improvement over state-of-the-art assays that often present false positives in patients who carry the gene but do not have an active CDI infection.

Conclusions:

The authors demonstrate an effective use of paper-based systems for gut bacteria detection that could almost certainly be expanded limitlessly to detect other new species. The sensors show suitable detection thresholds, almost on par with state-of-the-art qPCR methods when isothermal amplification is used, and high orthogonality. From a practical perspective, the assay is cheap, easy-to-use, and rapid (with readout in just a few hours). However, the claim that the assay is quantitative is muddier (see below). In addition, the novelty, given past work seems in paper sensors from the group, seems somewhat low. While the paper has an exciting take home message, some major issues would need to be addressed if it were to be published in Nature Communications.

Response:

We thank the reviewer for their overall positive view of our work. We address their specific comments below.

Major Issues:

- *The authors claim that in a previous paper, they demonstrate that "the toehold switch sensors exhibit a linear response to trigger RNA inputs in the low nanomolar to micromolar range." Yet in Figure 4b, following NASBA RNA amplification, the ON state varies with the logarithm of RNA*

concentration. An isothermal amplification strategy, if run to completion, would seem to make the final RNA concentrations closer together regardless of the starting seed concentration, which begs the question of how this experimental dependence originates. Can the authors make or justify a model that suggests that GFP production vs. starting RNA concentration before isothermal amplification should vary with a logarithmic dependence?

Response:

We thank the reviewer for this concern, however, we are not actually running NASBA reactions to completion for the express purpose of avoiding this problem. We apologize if this was unclear in our original text and we have revised the text to clarify this. We agree that if run to completion, the final RNA concentrations would be closer together regardless of the starting seed concentrations. Previous work using NASBA to quantify RNA concentrations have also shown log-linear relationships if reactions are not run to completion (Weusten et al., Nucleic Acids Res, 2002, 30, e26). As per Reviewer 1's suggestion, we did develop a mathematical model of the NASBA process (Supplementary Figure 7), which also confirms the log-linear relationship between input RNA concentrations and amplified RNA.

- The correlation curves developed in Figure 4b and Supplementary Figure 7 are plotted across very different concentration ranges and with a different number of points. Considering that many of these correlations appear poor, with large error bars, (particularly B. longum, which fits two parameters on three points), it does not seem fair to call this technique quantitative. This is particularly the case for Figures 4c and e, where the agreement between RT-qPCR and the predicted paper-based diagnostic diverge by factors of 3, 12, and 30, solely based upon the bacterial species and sample tested (spiked-in bacteria vs. clinical sample). If this technique cannot reliably predict the gold standard RT-qPCR reading within a factor of 5, then it cannot be called quantitative in my opinion. The positive results in Fig. 4d suggest that the word "semi-quantitative" is a fine alternative. Alternatively, if the authors can justify why RT-qPCR is underestimating the RNA concentration in the sample, this would be adequate.

Response:

We thank the reviewer for this critique. We are not able to match RT-qPCR copy number values within a factor of 5 and have thus changed the wording in the paper to "semi-quantitative".

Minor Issues:

- The authors claim that the in-house prepared extract behaves "suitabl[y] for our platform", yet the results in Fig. S12 suggest otherwise. The use of lacZ rather than GFP as a reporter is explained as "flexibility of the platform" - can the authors show results demonstrating that GFP is a sufficiently good reporter, considering the very low fold activation in this (enzyme) system compared to the complementary results in Fig. 3b? Otherwise, the authors should perhaps acknowledge that it does not work in the body of the manuscript.

Response:

We thank the reviewer for raising this concern. We have added results showing that the GFP reporter also works using in-house prepared extract. We apologize for not being clear in the presentation of our previous LacZ reporter data. We had reported our results in terms of endpoint absorbance, which is not comparable to our standard presentation of fluorescent protein production rate. We have re-plotted the data in terms of rate of change of absorbance. It is difficult to directly compare GFP production rate to the absorbance measurement since the

absorbance measurement is a combination of LacZ production and enzymatic conversion of CPRG. We acknowledge that the fold-activation using LacZ in the in-house prepared extract is low when compared to results using GFP in NEB PURExpress, however, the key point is that we are able to detect the 3 fM mRNA standard after NASBA and our reported sensitivity remains unchanged using this output.

- *Since the publication of SHERLOCKv2, please adapt the wording of the introduction's claim that "We demonstrated the utility of our platform in detecting...but we were not able to quantify their concentrations."*

Response:

We thank the reviewer for this concern, however, we consider our paper-based platform to be separate from the SHERLOCK platform. Therefore, our claim about quantification still holds true. However, we do recognize the importance of SHERLOCK and have added text to the Discussion of the revised paper to acknowledge it as a complementary technology.

- *The authors "mapped the primer locations to chemical structure probing data for E. coli 30S ribosomal subunits". Please include these data in the text.*

Response:

We thank the reviewer for this suggestion. We have added this information in a new Supplementary Figure 2.

Reviewer #2

Remarks to the Author:

Takahashi et. al. describe application of their toehold switch sensors and isothermal RNA amplification technique (NASBA) to detect bacterial transcripts in their manuscript, "On-demand analysis of the gut microbiome using paper-based RNA sensors". Two potential clinical diagnostic scenarios are described; the quantitative detection of marker RNAs from a panel of fecal bacteria (a surrogate for more extensive metagenomic studies) and the quantitative detection of C. difficile toxin mRNA in fecal samples.

The current manuscript builds upon the laboratory's prior exciting publications on toehold switch sensor and NASBA technology. Therefore the submission is not a proof-of-principle description of major new methodology per se. Yet, the elegant technology has been applied here to growing or well developed diagnostic targets, and as such hold great promise. Quantitative detection of C. difficile tcd gene abundance is likely to be among the most important applications of the RNA-targeted quantitation methodology. It is notable that the assays could also likely be adapted to detection of host transcripts in clinical material. Such dual host-microbial assays would be novel and likely to have enhanced utility over microbe-targeted detection alone. For example, detection of host inflammatory transcripts would likely help distinguish C. difficile colonization from active C. difficile-associated colitis. Further, simultaneous detection of host and bacterial transcripts applied to clinical metagenomics will likely have major utility in a number of disease states. The NASBA-toehold switch sensor methodology may be ideal in those capacities, given the low cost of the assay reagents.

Response:

We thank the reviewer for recognizing the unique ability of our platform to provide dual host-microbial detection. In light of the reviewer's suggestion, we have developed and added paper-based sensors for four host transcripts: calprotectin S100A9, CXCL5, IL-8, and oncostatin M (OSM). Calprotectin, CXCL5, and IL-8 are indicators of inflammation. Fecal calprotectin protein assays are already widely used in the diagnosis and monitoring of inflammatory bowel disease (IBD). Detection of OSM also has the potential for immediate clinical impact. A recent study found that high levels of OSM correlated with patients that did not respond to anti-tumor necrosis factor (TNF)-alpha therapies (West et al., *Nature Medicine*, 2017, 23 (5), 579.). We validated these sensors using clinical samples from patients with IBD and achieved good correlation with RT-qPCR. Additionally, recent studies have shown that host CXCL5 and IL-8 fecal mRNA has a far greater predictive accuracy for active *Clostridium difficile* infection than toxigenic bacterial burden (El Feghaly et al., *Clinical Infectious Diseases*, 2013, 56 (12), 1713; El Feghaly et al., *Journal of pediatrics*, 2013, 163 (6), 1697). We believe these additional experiments significantly enhance and clearly illustrate the potential clinical utility of our platform. We thank the reviewer for their valuable suggestion.

In addition to their tour de force application of RNA biology to clinical diagnostics, the manuscript describes several optimization steps which dramatically improve assay performance. In particular, the manuscript details optimization of the selectivity of various NASBA and sensor targets. Internal controls were developed and described which account for variability in signal strength and for background signal. Here, there are remaining concerns regarding assays performance. The authors describe significant continued run-to-run variability in signal response, as well as high background when run in clinical materials, and decreased sensitivity relative to qRT-PCR.

Response:

We thank the reviewer for raising these points. We acknowledge that there is run-to-run variability in our assay, however, we show that running a single control reaction alongside any unknown samples allowed us to correct for run-to-run variability. The control reaction allows us to use a pre-determined calibration curve to calculate mRNA concentration. We have revised the manuscript text to clarify this point.

In this demonstration of our platform, we did not observe any increase in background effects from the processed stool samples. It is true that for our Zika diagnostic, we observed an inhibitory effect of plasma/serum on the NASBA reactions, however, simply diluting the plasma/serum samples by 10-fold removed the inhibitory effect and still allowed us to detect 3 fM Zika RNA in plasma.

We acknowledge that we do not match the limit of detection of RT-qPCR; however, we believe that there are many applications where our sensitivity is sufficient. Further clinical studies must be done to determine the sensitivity necessary for each application; however, we have shown at least 3 fM sensitivity for all of our sensors, as low as 30 aM for some sensors in stool total RNA, and detection of both bacterial and host biomarkers from clinical stool samples.

Taken together, the data presented here suggest that the technology holds great promise but is not yet sufficiently robust to be applied as diagnostic assay for either of the two described scenarios. As improvements in assay performance continue to occur, these assays will indeed likely have important clinical utility, and there are likely to be even broader clinical applications of NASBA-toehold switch sensor technology yet to be described.

Response:

We thank the reviewer for the overall positive view of our work and their suggestions for improvement. We hope we have addressed their concerns over assay performance in our response above and our validation of both bacterial species identification and host inflammation markers against RT-qPCR using clinical stool samples. We have also broadened our discussion of the potential clinical uses in the text of the revised paper as suggested by the reviewer.

Reviewer #3**Remarks to the Author:**

The work by Takahashi et al describes the development of a paper-based RNA sensor system to quantify microbiota samples in a cheap and direct way. The authors leveraged a series of innovations previously published by their group that combined toe-hold sensors, cell-free reactions, and paper-based deployment and extends the method to measure individual microbiota species from synthetic and natural communities. The authors further develop orthogonal sensors that can detect each of 10 distinct target species without cross-talk signals. Finally, a demonstration of the system for use in detection of C. difficile levels is presented.

Overall, the manuscript is clear, succinct, and well written and the data supports the claims of the work. The methodology utilized in the study is not particularly novel as it has been previously published. Nonetheless, the application of the strategy to quantify gut microbiota samples is novel. While the reviewer appreciates the interesting demonstration of the system for detecting 10 model gut bacteria, the practical use of such a detection system for research or clinical application is somewhat of a stretch, at least in the context of quantifying commensal bacteria. In comparison to RT-PCR methods, the current approach (including use of NASBA for better signal detection) seems to take a little bit longer and requires running standards for each reaction run. The cost saving is marginal to RT-PCR methods. RT-PCR requires a dedicated instrument although the GFP-based toehold sensor still requires a quantitative UV-vis spectrophotometer for measurement. Both RT-PCR and toehold sensors are specific to the target species and thus individual primers and reaction conditions need to be developed. This of course is in contrast to Next-Gen Sequencing approaches which is more involved, but does provide much greater analytical power and throughput.

Response:

We thank the reviewer for raising these points. We agree that next-generation sequencing (NGS) provides greater analytical power and throughput, however, we believe that the combination of cost, time, and expertise required to run NGS hinders its use in clinical or low-resource settings. We use the panel of 10 bacteria to demonstrate the capabilities of our platform, but agree that currently there is no clinical application for quantifying these particular commensal bacteria. We envision that once researchers determine the subset of bacteria or RNA transcripts important for their study or application using NGS, our platform could be used as a cheaper and simpler alternative for collecting data. With regards to RT-qPCR, we agree that our assay takes the same amount of time and also requires similar upfront development to screen primers, etc.; however, our platform offers significant advantages over RT-qPCR. We only require running a single standard for quantification, where RT-qPCR requires a minimum of five (Svec et al., *Biomolecular Detection and Quantification*, 2015, 3, 9-16). When using commercial cell-free and NASBA reagents, our platform is ~10x cheaper than RT-qPCR. Furthermore, if in-house cell-free extracts and individually sourced NASBA reagents are used,

our platform is ~70x cheaper per transcript tested. This is a significant cost savings, especially for use in low-resource settings where our platform is especially suited. We have also shown that our platform works well with a colorimetric output (LacZ) that could be used with our previously published portable electronic reader. We have revised the text in the Discussion section to better highlight these important points.

All of those things said, the authors proposes a compelling application of their method to detect a clinical pathogen, which has fast turnaround, high sensitivity, and can distinguish between high and low toxin expressing variants. What can be achieved in this paper for clinical pathogens goes beyond current DNA-based methods. As such, it would seem that expansion of this study to other clinical pathogens such as CRE, VRE, etc. would further strengthen this work in a revised form. Further demonstrating and comparing the results with RNA toehold on clinical samples to detect different pathogens (and not just in a spike-in experiment of the clinical sample as demonstrated) and comparing with current RT-PCR based methods would show direct translational application to boost the study's impact. Overall, the work is promising and systematically presents a protocol to detect microbiota from complex samples using a single RNA-based toehold sensor system that is easy to deploy in resource limited environments.

Response:

We thank the reviewer for acknowledging the utility of our platform for detecting toxin mRNA. We agree that detection of CRE and VRE pathogens would be an interesting application of our platform especially in the differentiation of active infection versus colonization. However, despite our efforts in reaching out to local researchers and hospitals, we were not able to obtain appropriate clinical samples with *C. difficile*, let alone additional clinical samples for CRE and VRE. However, to further strengthen the clinical relevance of our platform, we followed up on a suggestion from Reviewer 2 and added a panel of sensors to detect host biomarkers of inflammation. We were able to obtain clinical samples from patients with inflammatory bowel disease and detected the host transcripts with good correlation to RT-qPCR.

Minor points:

1) *The authors should detail how the original toehold designs were generated for each of the species (i.e. Suppl Fig 1) and which ones were finally chosen for the subsequent studies (Fig 2b).*

Response:

We thank the reviewer for this suggestion. We have added more details to the Methods section about our design and screening process. We also highlighted the final sensor selection in Supplementary Figure 1.

2) *Some better quantitative analysis of the signal to noise (e.g. real signal vs background activity) could be applied or at least detailed. Currently, the rational for choosing what a good design is seems to be somewhat qualitatively. Quantitative treatment could help guide future users to generate suitable and high-performing toehold designs.*

Response:

We thank the reviewer for this suggestion. We have added fold-activation data to Supplementary Figure 1 for more quantitative analysis of the signal to noise. We have also described the 5-fold activation requirement we used to define the minimum functional threshold for all our published toehold sensors.

REVIEWERS' COMMENTS:

Reviewer #1 (Remarks to the Author):

The authors have addressed my main concerns and I support publication. It is an exciting piece.

Reviewer #2 (Remarks to the Author):

The authors have done a nice job responding to the reviewer's comments. The addition of host transcript analysis strengthens the manuscript considerably. The new figures also strengthen the manuscript; they help the reader follow the protocol and data interpretation.

Reviewer Comments

Reviewer #1

Remarks to Author:

The authors have addressed my main concerns and I support publication. It is an exciting piece.

Response:

We thank the reviewer for reviewing our updated manuscript. We are grateful for their comments and are pleased that we addressed all their concerns.

Reviewer #2

Remarks to Author:

The authors have done a nice job responding to the reviewer's comments. The addition of host transcript analysis strengthens the manuscript considerably. The new figures also strengthen the manuscript; they help the reader follow the protocol and data interpretation.

Response:

We thank the reviewer for reviewing our updated manuscript. We are grateful for their comments and thank them for their support.